# Enhanced Public Interest in Response to the Refugee and Healthcare Crises in Greece

**DOI:** 10.3390/ijerph17072272

**Published:** 2020-03-27

**Authors:** Ourania S. Kotsiou, Vaios S. Kotsios, Konstantinos I. Gourgoulianis

**Affiliations:** 1Department of Respiratory Medicine, Faculty of Medicine, University of Thessaly, BIOPOLIS, 41110 Larissa, Greece; kgourg@med.uth.gr; 2Metsovion Interdisciplinary Research Center, National Technical University of Athens, 44200 Athens, Greece; vaioskotsios@gmail.com

**Keywords:** economic crisis, Greece, Google Trends, health, public interest, refugee

## Abstract

*Background*: The Greek National Health System (NHS) has been profoundly affected by the synergy of the economic and refugee crises. We aimed at evaluating the public interest regarding refugee and healthcare issues in Greece. *Methods*: Google Trends was employed to normalize traffic data on a scale from 0 to 100, presented as monthly relative search volume (RSV) for the search term queries: “refugees”, “health”, “diseases”, “hospital”, and “economic crisis” in Greece, from the period 2008 to 2020. Cross-country comparisons in selected European countries were made. *Results*: The analysis of RSV data showed an upward trend for the keyword “refugee”, in Greece, in the last five years, with two remarkable peaks from 2015 to 2016 and from 2019 to the present. Interest regarding refugees was more prevalent in the Aegean islands compared to the mainland. The mass influx of refugees has been linked to disease-related concerns. The search terms “hospital” and “health” have been the most popular and constantly quested topics since the beginning of the economic crisis in Greece, in 2009. Similar trends existed across Europe. *Conclusion*: There is an urgent need for effective public awareness of current politico-ethical and social-economic conditions. The patterns of public interest can formulate public policy.

## 1. Introduction

Greece is called the country of “Xenios Zeus”, the Ancient Greek god of foreigners and hospitality [1]. However, Greece has been grappling with a severe socio-economic crisis since 2009 [1]. From the period 2015–2016, Greece experienced an unprecedented influx of refugees and migrants fleeing their home countries in the Middle East because of war, which created what is now known as the most massive displacement of a people since World War II [1,2]. The rapid entry of refugees into Greece raised the critical issue of health policy [1,2]. The Greek National Health System (NHS) has been profoundly affected by the synergy of the economic and refugee crises leading to unmet needs for health care and subsequent adverse health outcomes for the local and refugee population [1,2].

The Eastern Greek islands have constituted direct passageways of refugees to the European continent in recent years. Public health facilities in the Aegean islands have been significantly hit by the long-lasting economic crisis [1,2]. They have been overstretched even more with the demands of the unprecedented influx of refugees and appeared inadequately prepared to address both locals’ and refugees’ needs [1,2].

Overcrowded reception centers and hotspots have been associated with severe disease burden [1,2]. The forced migration, malnutrition, questionable hygienic living conditions, legal insecurity, financial and social isolation, racism, communication barriers, and employment difficulties place refugees in jeopardy of mental and physical illness [1,2]. On the other hand, the economic crisis has forced a million Greeks back into poverty. The locals have unmet needs due to the cost or long waiting times of medical examination or treatment in public health facilities [1,2]. Understanding the health needs and concerns of the local and refugee population is essential for the prevention of racism and discrimination.

Similarly, Europe has been gripped in a political and humanitarian crisis, as the incoming numbers overwhelmed both individual state and collective European Union ability to respond effectively [3]. How these frames affect public opinion is an emerging issue. The humanitarian refugee crisis has led to heated debates at the political level and in the media, as well as to deep concerns among the European public at large.

To better understand the general public agony, it is of high importance to evaluate public interest in getting information about the refugee crisis, health issues, and the NHS in Greece, making cross-country comparisons with other European countries that were also severely hit by the crises. We aimed to evaluate the public interest in these major topics by using Google Trends data.

## 2. Materials and Methods

Google Trends is a website by Google that analyzes the popularity of the top search queries in Google Searches across various regions and languages [4]. In our study, Google Trends was employed to normalize traffic data on a scale from 0 (<1% of the peak volume) to 100 (peak of traffic), presented as monthly relative search volume (RSV), concerning “refugee”, “health”, “hospital” and/or “diseases” and “tuberculosis” as search terms, in Greece [5]. The RSV data were filtered by geographic regions in Greece. Furthermore, the RSV trends of the Google searches for the keywords “refugees”, “health”, “hospital”, and “economic crisis” were collected to investigate cross-country comparisons across eight European countries that were severely affected by the crises. The time-series analysis covered the last 12 years (January 2008 to March 2020). Google trends data were further analyzed and visualized using Business Intelligence Analysis tools (Tableau Software LLC, Seattle, WA, USA).

## 3. Results

The analysis of search trends data regarding the refugee crisis showed an upward trend about this issue in Greece in the last five years (Figure 1).

Specifically, there was a remarkable peak for the search term “refugee” from 2015 to 2016. Volume searches were leveraged up, reaching 70% in March 2016, above annual means. A fast downward RSV along the June–July summer months of 2016 was also noteworthy. This topic has remained a hot topic for search along the remaining period, as was also evidenced by a second peak element in November 2019 up to the present time (March 2020).

The RSV data were filtered by geographic regions in Greece. The interest regarding refugees was more prevalent in the main reception centers, such as in the islands of Lesvos, Chios, Kos compared to the mainland (Figure 2). Notably, since 2015, the word “refugee” has become a more popular search term than the search term “tuberculosis”, in Greece.

Overcrowded reception centers and hotspots are highly demanding and are associated with severe disease burden. Notably, Google trends show that the mass influx of refugees in Greece in 2016 was linked to public disease-related concerns thereafter (Figure 3).

The search terms “hospital” and “health” have been the most popular and constantly quested topics since the beginning of the economic crisis in Greece (Figure 1), especially on the mainland (Figure 2). The volume of searches for the keyword “hospital” reached 100%, which was the peak popularity for the term, at the beginning of the economic crisis in 2009 (Figure 1).

Similar time-dependent patterns existed in the Google search trends for the keywords “refugee”, “economic crisis”, “health”, and “hospital” among other European countries that were severely affected by the synergy of economic and refugee crises, as presented in Figure 4.

## 4. Discussion

The increasing number of searches for community-related issues generates “big data,” providing meaningful research in infodemiology, which is the study of patterns and determinants of information on the Web to inform public health and public policy. Google Trends, which is characterized by its low cost, transparency, simplicity, and reproducibility across a variety of domains, provided essential data regarding public interest for several current social topics. Enhanced public interest in response to the refugee, economic and healthcare crises in Greece has been detected from the analysis. Similar time-dependent patterns existed across Europe regarding these major issues.

The unprecedented rise in the number of asylum seekers and migrants entering the country has been linked to economic and social consequences for health outcomes. Our findings show that there are signs of awakening public interest and concern about those significant social challenges. The volume of searches regarding the term “hospital” remained a hot topic since the beginning of the economic crisis. The crisis negatively affected health expenditure in Greece and the health status of the population. Total health expenditure in the country fell from €22.49 billion in 2009 to €14.73 billion in 2015, while the mean per capita healthcare expenditure declined from €2024 in 2009 to €1361 in 2015, an overall decrease of almost 33% [6,7]. According to estimates made by Eurostat in 2015, 35.7% of the population was at risk of poverty or social exclusion [8]. Consequently, in the years of downturn, there has been a shift from the private to the public healthcare sector, as shown by an increase of 24% in the number of admissions to public hospitals from the very beginning of the crisis (from 2009 to 2010) which continued to rise in the first half of 2011 by 8% [6]. Conversely, a decrease in admissions to private hospitals in the period 2009–2010 has been documented [2].

A further burden put on to the already heavily-loaded Greek health system comes from the thousands of refugees who pass the Greek borders. The Greek reception system, especially in the Eastern Greek islands, has been strongly criticized as inadequate by Greek civil society as well as national and international non-governmental organizations. Nowadays, flaming tensions simmer between refugees and locals at overcrowded Greek islands [9]. In a tense political climate, public opinion claims that the rights of refugees to secure and hygienic living conditions, and of island residents to exist, to live and create without deteriorating the quality of their daily lives, should be strictly protected [9].

In addition to non-communicable diseases, communicable diseases account for a significant morbidity burden in newly arrived migrants and refugees. The high disease burden of refugees in Greece in 2016 has been connected to broader public disease-related concerns, as shown by our analysis.

The analysis shows that the public interest in the European countries most affected by economic and refugee crises followed a similar distribution pattern of RSV searches on refugee, economic, and healthcare issues. Indeed, large groups of European citizens expressed humanitarian concerns, and pointed at the moral duty to help people in need [9]. Large groups of other citizens, however, were worried about the threat that Islamic refugees would bring to their own culture and safety [9].

## 5. Conclusions

Besides being a humanitarian disaster, the refugee crisis posed political, healthcare, and societal challenges across the European Union. The synergy of the economic and refugee crisis also sparked the large-scale interest of concerned citizens, as shown in novel Internet-based evidence. The increasing number of searches for community-related topics generates “big data,” providing meaningful research in infodemiology. Information dissemination via the Internet is necessary for increasing clarity around the growing migratory, economic, and healthcare challenges the world is facing and is proving to be one of the most useful data-driven tools for tackling the refugee, socio-economic and health challenges [10]. In other words, these data are essential to understanding the patterns and determinants of public interest in social issues, helping to formulate public policy further. Societies differ in how they interpret the specific content of public interest. Undoubtedly, these data revealed the urgent need to create effective social awareness of current political, ethical, economic, and social conditions. When all the voices of society are considered iregarding an issue, it is essential for the government to actually listen to the people.

## Figures and Tables

**Figure 1 ijerph-17-02272-f001:**
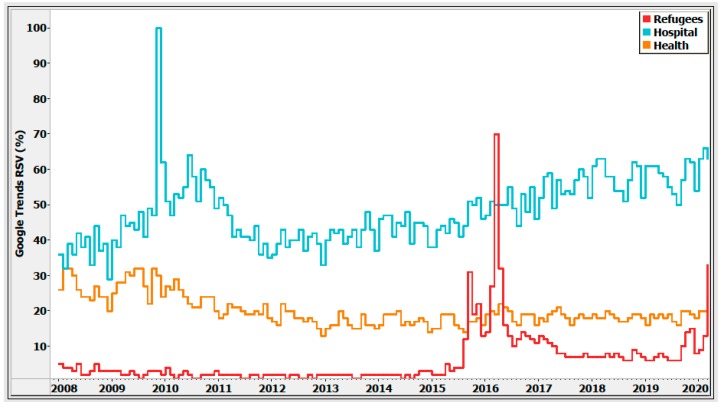
Relative search volume (RSV) for current social issues in Greece (search terms: “refugees”, “health”, “hospital”), January 2008–March 2020. Note: Search volume data provided by Google Trends (https://www.google.com/trends) [4].

**Figure 2 ijerph-17-02272-f002:**
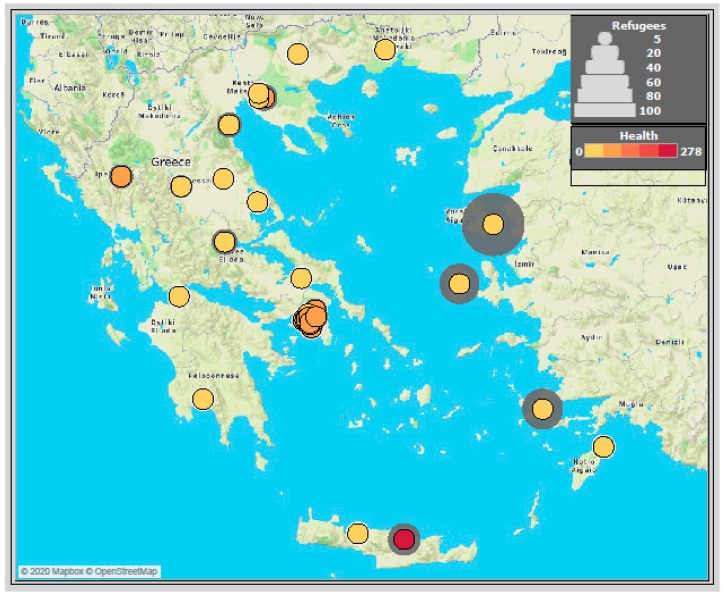
Spatial map of Google Trends based on refugee or health-related data, Greece (January 2008–March 2020). Note: The figure is based on the map provided by Google Trends. The color gradient and the size of the circle correlate with the volume of “health” or “refugees”-related Web searches in Greece, respectively.

**Figure 3 ijerph-17-02272-f003:**
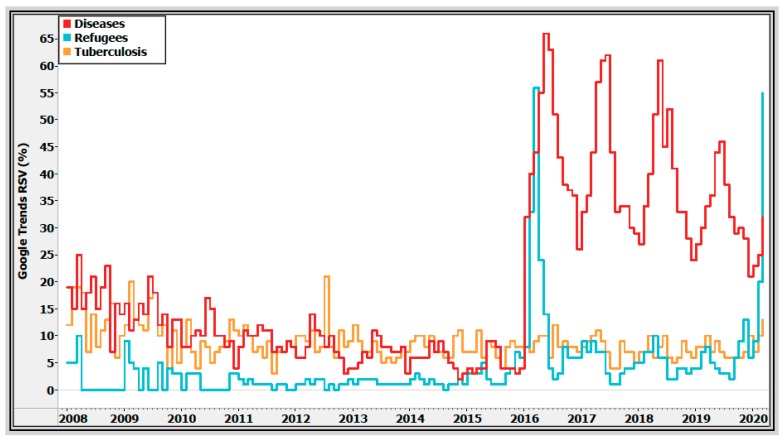
Google Search Trends for the search terms queries “refugees”, “diseases”, and “tuberculosis” in Greece, January 2008–March 2020. Note: Search volume data provided by Google Trends (https://www.google.com/trends) [4].

**Figure 4 ijerph-17-02272-f004:**
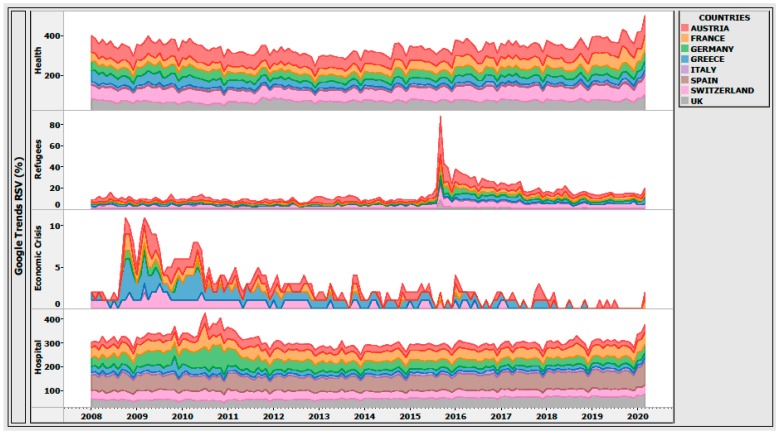
Comparison between relative search volume (RSV) trends of the Google searches for the keywords: “refugee”, “economic crisis”, “health”, and “hospital” between the eight most-affected European countries by the economic and refugee crises, January 2008–March 2020. The RSV volume is indicated by the thickness of each line graph. Note: Search volume data provided by Google Trends (https://www.google.com/trends) [4].

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
