# Peer review of "Enhanced Public Interest in Response to the Refugee and Healthcare Crises in Greece"

_ijerph, 2020, doi:10.3390/ijerph17072272_

Round 1

Reviewer 1 Report

Very interesting approach and data, would be intersting to do this with other european countries according to economic Level and refugee population

Maybe the authors could give more details about the situation in the " hotspots" ( blue on their map)?
Another suggestion would be to expand the search for more search words in the refugee context: medication, waiting times hospital or camp capacity.
A relation to of the search terms to political decisions would be interesting as well.

Author Response

COMMENTS FROM REVIEWER 1:

  1. Very interesting approach and data

RESPONSE: We sincerely thank you for your kind words about our paper. We are delighted to receive positive feedback from you. We appreciate you taking the time to offer us your insights related to the manuscript. In the following sections, you will find our responses to each of your points and suggestions.

  1. would be interesting to do this with other European countries according to economic Level and refugee population.

RESPONSE: Thank you for raising this important point regarding the extension of our analysis to the European Union. Per your suggestion, the relative search volume (RSV) Google trends for the keywords: “refugee”, “health”, “hospital” and “economic crisis” related to economic, refugee and healthcare issues were collected to investigate cross-country comparisons between the most-affected European countries by the crises, from January 2008 to March 2020 (Figure 4). Notably, similar time-dependent patterns concerning those search terms existed among European countries. In the revision, we developed two paragraphs highlighting this topic on the Result (Page 4, lines 106-108) and Discussion sections (Page 5, lines 150-155), respectively. Accordingly, we have reorganized the abstract according to the proposed approach (Page 1, lines: 16, 22). Finally, Figure 1 has been updated and visualized by a Business Intelligence Analysis tool to show the new data up to March 2013 in Greece.

  1. Maybe the authors could give more details about the situation in the "hotspots" (blue on their map)?

RESPONSE: Thank you for this valuable comment. We agree that this point needed to be expressed in more detail. In the revision, we discuss the situation in the “hotspots” in-depth, found on Pages 1-2, lines 37-49 and Page 5, lines 138-149. Accordingly, Figure 2 has been updated to show the RSV data up to March 2020, in Greece. In the revision, it is visualized by a Business Intelligence Analysis tool. Furthermore, the most detailed analysis shows that the high influx of refugees in 2016, has been linked to broader disease-related concerns thereafter, as shown in Figure 3, and reported in Page 3, Lines 94-96 and Page 5, Lines 146-149.

  1. Another suggestion would be to expand the search for more search words in the refugee context: medication, waiting times hospital or camp capacity. A relation to the search terms to political decisions would be interesting as well.

RESPONSE: Thank you for this direction. Although the proposed analyses are important, they are not easy to be performed as, unfortunately, Google Trends revealed that these search terms or other relevant keywords (such as emergency/hospital wait/waiting times, hot-spot/reception centers, overcrowding, politics, political measures/meters, laws, government, etc.) were not somehow popular search topics in Greece. Data were not sufficient for further analysis.

Thank you for your very careful review of our paper, and for the comments and suggestions that ensued. We are grateful for the time and energy you expended on our behalf. We hope that you find these revisions an improvement.

Reviewer 2 Report

The brief report is interesting. However, I suggest the authors strengthen the introduction, material and methods and conclusions. These sections need to be strengthened.

Author Response

COMMENTS FROM REVIEWER 2:

  1. The brief report is interesting. However, I suggest the authors strengthen the introduction, material and methods and conclusions. These sections need to be strengthened.

RESPONSE: We appreciate you taking the time to offer us your insights related to the paper. We are delighted to hear that you read our work with interest. We agree that these sections needed to be strengthened. Per your advice, we have re-written most of the Introduction section, (Pages 1-2, lines 37-54), and Conclusions sections (Pages 5-6, lines 157-169). We now discuss in more detail the impact of refugee and healthcare crises in the Greek reception centers as well as across the European Union. We have revised the Material and Methods section (Page 2, lines 65-70) according to the new approach.

Besides, we have extended our analysis to the European Union. In the revised paper, we have added Figure 4 to visualize comparisons between relative search volume (RSV) trends of the Google searches for the keywords: “refugee”, “health”, “hospital”, and “economic crisis” between eight European countries that were severely affected by the economic and refugee crises (January 2008 - March 2020).

Furthermore, we have added Figure 3 to show that the high influx of refugees in Greece, in 2016 has been linked to broader disease-related concerns thereafter. Finally, Figures 1 and 2 have been updated to show the current data, up to March 2020, by a Business Intelligence Analysis tool. We hope that you find these revisions an improvement.

We found your feedback very constructive. We tried to be responsive to your concerns.

Reviewer 3 Report

The manuscript should also consider a more global dimension, it seems to me too limited to the area of Greece.
The authors should give a more global framework both in the introduction and in the conclusions, trying to broaden the practical implications of their work by extending it to other contexts

Author Response

COMMENTS FROM REVIEWER 3:

  1. The manuscript should also consider a more global dimension, it seems to me too limited to the area of Greece. The authors should give a more global framework both in the introduction and in the conclusions, trying to broaden the practical implications of their work by extending it to other contexts.

RESPONSE: We thank you for taking the time and energy to help us improve this paper. Thank you for this valuable direction. We agree that we should give a more global dimension in our analysis, and we apologize for any omission. We extended our work to other European countries with various economic status affected by the refugee crisis. Per your advice, we have revised the Introduction (Pages 1-2, lines 37-54), and Conclusions sections (Page 5-6, lines 157-169) along the lines you proposed.

Accordingly, we have added Figure 4 to visualize comparisons between relative search volume (RSV) trends of the Google searches for the keywords: “refugee”, “economic crisis”, “health” and “hospital” between the eight most-affected European countries by the economic and refugee crises for the period: 2008-2020.

Furthermore, we have added Figure 3 to show that the high influx of refugees in Greece, in 2016 has been linked to broader disease-related concerns thereafter.

Finally, Figures 1 and 2 have been updated to show the current data up to March 2020 by a Business Intelligence Analysis tool. We hope that you find these revisions an improvement.

We found your feedback very constructive. We are grateful for the time and energy you expended on our behalf. We hope that you find these revisions an improvement.

Round 2

Reviewer 3 Report

the authors reviewed the article according to the indications. It's OK for me